# The Possible Effects of Zinc Supplementation on Postpartum Depression and Anemia

**DOI:** 10.3390/medicina58060731

**Published:** 2022-05-29

**Authors:** Chieko Aoki, Kenji Imai, Taro Owaki, Tomoko Kobayashi-Nakano, Takafumi Ushida, Yukako Iitani, Noriyuki Nakamura, Hiroaki Kajiyama, Tomomi Kotani

**Affiliations:** Department of Obstetrics and Gynecology, Nagoya University Graduate School of Medicine, 65 Tsurumai-Cho, Showa-Ku, Nagoya 466-8550, Japan; ycey166324@med.nagoya-u.ac.jp (C.A.); tarosakujiro@gmail.com (T.O.); tnakano@med.nagoya-u.ac.jp (T.K.-N.); u-taka23@med.nagoya-u.ac.jp (T.U.); yukakoi@med.nagoya-u.ac.jp (Y.I.); nakamura-n@med.nagoya-u.ac.jp (N.N.); kajiyama@med.nagoya-u.ac.jp (H.K.); itoto@med.nagoya-u.ac.jp (T.K.)

**Keywords:** EPDS, Zn, Fe, obstetrics, cesarean section

## Abstract

*Background and Objectives:* The effects of postpartum zinc supplementation are still unclear. Our purpose in this study is to investigate the association between Zn supplementation and postpartum depression, defined by an Edinburgh Postnatal Depression Scale (EPDS) score ≥ 9, and the effect on the hematological status of postpartum women. *Materials and Methods:* We first investigated whether zinc supplementation affected the perioperative levels of zinc, hemoglobin, and hematocrit in 197 cases who underwent cesarean section and had postpartum anemia. Next, logistic regression analyses were performed on 148 eligible cases to determine the association between zinc supplementation and postpartum depression. *Results:* Postpartum zinc supplementation significantly improved the status of maternal blood zinc levels and reduced the risk of developing postpartum depression (adjusted odds ratio: 0.249; 95% confidence interval: 0.062–0.988; *p* = 0.048). Iron supplementation is a standard and effective strategy for treating anemia; however, the combination of oral iron plus zinc supplementation resulted in slightly significant negative effects on postpartum hemoglobin and hematocrit compared to oral iron supplementation only. *Conclusions:* Postpartum zinc supplementation causes a significant positive effect on postpartum depression (EPDS score ≥ 9). Zinc supplementation had a negative but transient influence on the hematological status in women with postpartum anemia treated with oral iron supplementation; however, the differences were not clinically significant. Thus, we did not regard it as an adverse effect to be considered, and postpartum zinc supplementation may be viewed as beneficial in postpartum women.

## 1. Introduction

Zinc (Zn), the second most abundant trace element in the human body, is a well-known essential co-factor of more than 300 enzymes that regulate various cellular processes and signaling [1]. Diet is the main source of Zn [2]. Although the daily requirement of Zn during pregnancy is increased due to the fetus and placenta [3], most pregnant women do not achieve the recommended daily intake [4,5]. Maternal Zn deficiency during pregnancy can be a risk factor for adverse outcomes such as anemia, preterm birth, hypertensive disorders in pregnancy, low birth weight, and postpartum complications [3,6]. However, scientific data on the influence of Zn supplementation on the course of pregnancy are divergent [1]. A recent Cochrane review did not show the benefits of Zn supplementation; antepartum Zn supplementation did not affect pregnancy outcomes, including preterm birth, hypertensive disorder in pregnancy, and neonatal outcomes [7]. Since the benefits have not been demonstrated, antepartum Zn supplementation has not yet been recommended [8]. On the other hand, the effects of postpartum zinc supplementation were not mentioned in the Cochrane review. The effects of postpartum Zn supplementation on postpartum anemia and postpartum depression are still unclear.

Postpartum anemia is one of the most prevalent medical disorders seen during pregnancy. Iron (Fe) deficiency is a common cause of anemia; thus, Fe supplementation is frequently recommended. Zn status is also associated with hematological abnormalities in pregnancy [9]. However, there is no clear consensus on the efficacy of Zn supplementation for antepartum anemia, which could be explained to some extent by the competition of Zn and Fe for absorption by the intestines [9,10,11]. Moreover, few studies have examined the effects of Zn supplementation on postpartum anemia.

Postpartum anemia and nutritional deficiencies, including low levels of Zn, are known risk factors for postpartum depression, a serious mental illness with detrimental impacts on the mother and child [12,13,14]. Low intake of Zn is associated with an increased rate of depression among pregnant women [15]. Decreased maternal Zn levels have been observed in cases of postpartum depression. However, there are few data on the effects of Zn supplementation on postpartum depression.

Therefore, our purpose in this study was to investigate the association between Zn supplementation and postpartum depression, defined by an Edinburgh Postnatal Depression Scale (EPDS) score ≥ 9, and the effect on the hematological status of postpartum women. 

## 2. Materials and Methods

### 2.1. Study Population and Clinical Data

The present study was a retrospective case–control study, and it was approved by the Institutional Review Board of Nagoya University (approval number 2015-0415; approval date 4 March 2016). All of the clinical data on maternal and neonatal characteristics were obtained from the hospital records. Nagoya University Hospital is a tertiary care hospital that manages high-risk pregnancies, with routine monitoring of the maternal blood Zn levels in the perioperative period of cesarean section (CS). Because our study design was a retrospective case–control study, Zn supplementation or not was left to the discretion of each doctor.

As shown in Figure 1, the 382 cases of CS were performed in our institution between October 2019 and November 2021. To focus on the influence of Zn supplementation on postpartum anemia, we excluded the cases that did not have anemia. We also excluded cases with complications, such as autoimmune disorders or massive postpartum hemorrhage requiring blood transfusion, and cases with an incomplete medical record. Consequently, the first analysis (Analysis 1) was performed using the data of 197 enrolled cases. Moreover, the influence of Zn supplementation on EPDS was evaluated using the data of 148 cases after excluding the cases with the neonate admitted to the neonatal intensive care unit (NICU), neonatal deaths, and no EPDS data (Analysis 2). The EPDS is a self-administered questionnaire consisting of 10 items, and the onset of postpartum depression was assessed using the EPDS (Appendix A). In previous validations, postpartum depression could be detected in the first month postpartum with a cutoff value ≥9, showing good sensitivity and specificity [13].

### 2.2. Statistical Analysis

All of the data were analyzed using SPSS version 28.0 for Windows (SPSS, Inc., Chicago, IL, USA). In both Analysis 1 and 2, the maternal and neonatal characteristics were compared using the Fisher exact test for categorical variables and the Mann–Whitney U test for continuous variables. In Analysis 2, univariate and multivariate logistic regression analyses were performed to determine the association between Zn supplementation and EPDS. The crude and adjusted odds ratio (OR) and 95% confidence interval (CI) were estimated after the adjustment for maternal age at delivery, parity, infertility treatment, body mass index at delivery, and gestational age at delivery. A *p*-value of < 0.05 was considered statistically significant.

## 3. Results

The upper part of Table 1 shows the maternal characteristics of all the 197 cases enrolled in Analysis 1 (Zn supplementation, 82 cases; No Zn supplementation, 115 cases). Since the present study included 21 cases of twin pregnancy, the neonatal characteristics of the 218 cases (Zn supplementation, 93 cases; No Zn supplementation, 125 cases) are shown in the lower part of Table 1. Because we focused on the cases with postpartum anemia, Fe treatment was administered to all the 197 cases. Compared to the no Zn supplementation group, the percentage of cases that received intravenous Fe treatment was significantly higher in the Zn supplementation group, which may be due to the increased amount of blood loss during operation. Although a difference in cord blood pH was observed, the difference was not clinically significant.

The perioperative trend in maternal Zn levels is shown in Figure 2A. According to previous reports, Zn deficiency was defined as a Zn level of <60 μg/dL; marginal Zn deficiency as a Zn level of ≥60 and <80 μg/dL [13,16]. All the cases in the present study had Zn deficiency or marginal Zn deficiency before the operation, which further decreased on postoperative day 1. In most of the cases that received Zn supplementation, Zn acetate hydrate (with each tablet containing 50 mg of Zn; Nobelpharma Co, Ltd., Tokyo, Japan) was administered orally twice daily (100 mg/day) for 4 days from postoperative day 1. The Zn concentration in each case that received the Zn supplementation significantly increased on postoperative day 6. Moreover, none of the cases showed excess levels of Zn.

Next, we conducted a subgroup analysis based on the method of Fe administration. Figure 2B–E show the perioperative trends of maternal hemoglobin (Hb) and hematocrit (Hct) (Figure 1, Analysis 1). Fe supplementation is a standard and effective strategy for treating anemia; however, the combination of oral Fe plus Zn supplementation resulted in slightly significant negative effects on postoperative day 6 compared to oral Fe supplementation only (Figure 2B,C). The difference resolved at one-month postpartum, and such negative effects of Zn were not observed among the cases that received intravenous Fe supplementation.

The EPDS is the most widely accepted screening scale in the perinatal period and is used routinely for the first-month postpartum check-up in our country. Cases with an EPDS score of ≥9 are often considered to have postpartum depression or probable postpartum depression [13,17]. Thus, we examined the relationship between Zn supplementation and an EPDS score of ≥9 in 148 cases (Figure 1, Analysis 2). The baseline characteristics of the cases enrolled in Analysis 2 are shown in Table 2. A statistical difference was found in cord blood pH, but this difference is not considered to be clinically significant. We performed univariate and multivariate logistic regression analyses and found that maternal age at delivery (adjusted OR: 0.876; 95% CI: 0.780–0.984; *p* = 0.025) and Zn supplementation (adjusted OR 0.249; 95% CI 0.062–0.988; *p* = 0.048) were independently associated with an EPDS score of ≥9 (Table 3). To check the interactions between Fe and Zn supplementation, we conducted a subgroup analysis based on the method of Fe administration using the same method as in Analysis 1 (intravenous or oral administration) and found no trend or no significant differences in EPDS score of ≥9 (data not shown).

## 4. Discussion

Adequate maternal nutrition in pregnancy is imperative for the health of the mother and child [18,19]. Our findings suggest a clinical implication of postpartum Zn supplementation; elevated maternal Zn concentration caused by postpartum Zn supplementation may reduce the risk of developing postpartum depression, whereas the results of this study indicate that oral administration of the combination of Fe and Zn may transiently interfere with the hematological status, and thus care is required. To the best of our knowledge, this is the first report regarding the positive effect of postpartum Zn supplementation on postpartum depression in clinical settings.

Postpartum depression is a significant public health problem, affecting approximately 10–20% of postpartum women [20], and can lead to serious conditions such as suicidal attempts [21,22]. Deficiencies in trace elements during the postpartum period have been recognized as important contributors to postpartum depression [23]. Low serum Zn level is the hallmark of depression and treatment-resistant depression [24,25]. These previous reports support our findings that postpartum Zn supplementation has a positive effect on postpartum depression.

Various scientific data on the possible benefits of zinc supplementation during pregnancy have been reported—Zn supplementation minimized the risk of preterm birth and infection in newborns [26], was beneficial to fetal neurobehavioral development [27], and improved fetal well-being [28]—whereas there are only a few reports on the effects of postpartum Zn supplementation, particularly in relation to postpartum depression. Fard et al. reported a randomized controlled trial (RCT) studying the association between Zn supplementation and postpartum depression, in which postpartum Zn supplementation did not improve maternal depressive symptoms [29]. However, the dosage of Zn was very low (27 mg/day), and Zn levels after supplementation were not measured. Another RCT with non-postpartum patients reported that 25 mg/day of Zn supplementation could not elevate serum Zn levels. In our study, almost all patients received 100 mg/day for 4 days, and the maternal Zn levels were significantly elevated. Thus, this inconsistency between the RCT findings of Fard et al. and our results may be explained by the Zn dosage to some extent. Nikseresht et al. demonstrated that acute administration of combined treatment with Zn, magnesium, and vitamin B1 on postpartum day 3 significantly improved postpartum depressive symptoms in a mouse model [30], which also leads us to speculate that the administration method of Zn supplementation has a significant impact on postpartum depression.

Although the exact mechanisms of the benefits of Zn supplementation remain unknown, the immunomodulatory and neurological effects of Zn could be speculated upon; inverse associations were reported between a low Zn level and an increased CD4+/CD8+ T cell ratio [24,31], which could contribute to the clinical symptomatology and neuro-progressive pathways in depression. Zn has the potential to modulate the excitability of neurons by affecting the glutamate and gamma aminobutyric acid receptors [32]. Except for excess concentrations, the toxic effects of Zn supplementation have not been identified [8]. Thus, it may be prudent to include Zn supplementation in postpartum women.

Because of the known competition between Zn and Fe [11,33,34], we investigated whether the addition of Zn to Fe supplementation affected the hematological status. We showed that the combination of oral Fe plus Zn supplementation resulted in slightly significant negative effects on the levels of Hb and Hct on postoperative day 6 compared to oral Fe supplementation only, and such a negative effect was not observed among the group that received intravenous Fe supplementation. Although paying a lot of attention to this is not desirable, our results indicate that it may be disadvantageous to administer the combination of oral Fe and Zn in the postpartum period or that a sufficient interval between Fe and Zn administration should be observed.

This study has some limitations. Due to the small sample size, our findings might be statistically limited, and thus the results should be interpreted with caution. Since our analysis was performed only with patients undergoing CS, it is unclear whether the same effect would be seen in patients undergoing vaginal delivery. Additionally, we studied only the effect of Zn supplementation on cases who received Fe treatment simultaneously, which could be highly biased. Furthermore, the decision to administer postpartum Zn supplementation was doctor-dependent in the present study, which is an intrinsic limitation of the retrospective study design. Since depression may develop after the first month postpartum, it would be ideal to conduct a long-term follow-up (e.g., one year postpartum) to verify the effect of postpartum Zn supplementation. This is a retrospective study, and selection bias is inevitable to some degree; thus we are now planning a larger prospective study to conduct a more detailed analysis of these findings.

## 5. Conclusions

We reported that postpartum Zn supplementation significantly improved the status of maternal blood Zn levels and caused a significant positive effect on postpartum depression (EPDS score ≥ 9). Our data indicate that postpartum Zn supplementation has the potential to prevent postpartum depression. Zn supplementation had a negative but transient influence on the hematological status in women with postpartum anemia treated with oral Fe supplementation; however, we did not regard it as an adverse effect to be considered. This study was retrospective and had a small sample size, and thus the consideration of confounders was limited, meaning that our findings regarding Zn supplementation may be a coincidence. Therefore, given the complications of postpartum Zn deficiency, Zn supplementation may be considered in postpartum women, although prospective multicenter studies are required to confirm our findings.

## Figures and Tables

**Figure 1 medicina-58-00731-f001:**
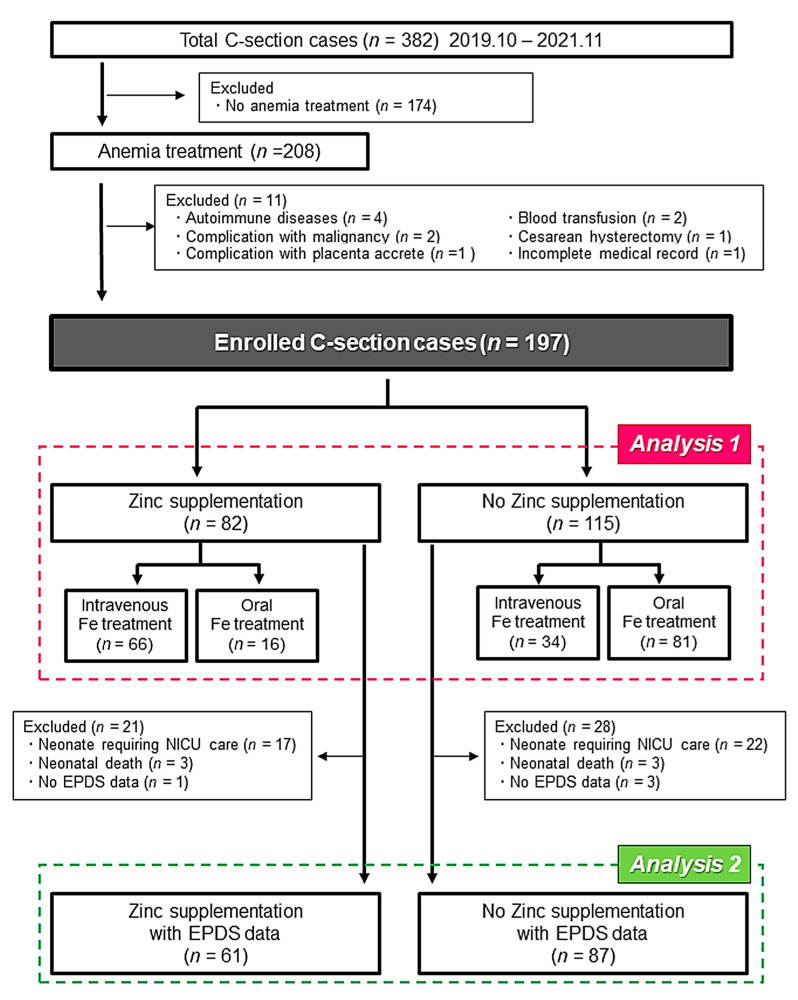
Flow diagram of the present study subjects. C-section, Cesarean section.

**Figure 2 medicina-58-00731-f002:**
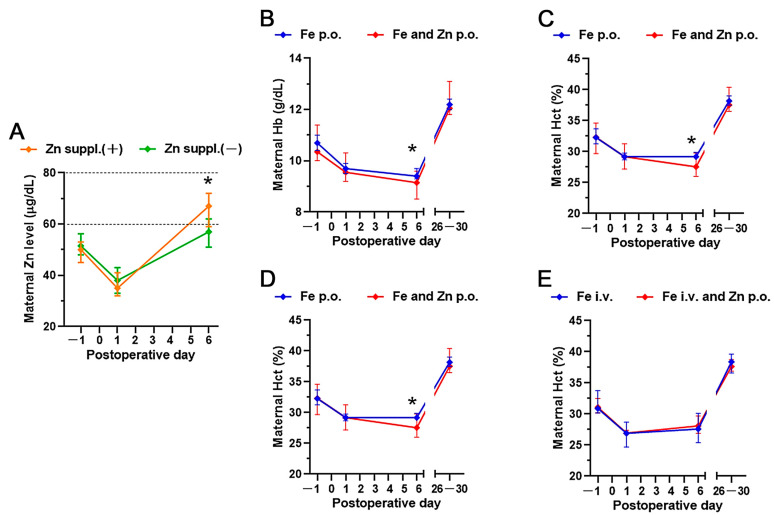
Perioperative trends of zinc, hemoglobin, and hematocrit after zinc supplementation or not. The median values of (**A**) maternal serum zinc, (**B**,**D**) hemoglobin, and (**C**,**E**) hematocrit are displayed (error bar: 95% confidence interval). The data were analyzed using the Mann–Whitney U-test on each evaluation day, and the asterisk means *p* < 0.05. suppl, supplementation; p.o., per os; i.v., intravenous medications; Hb, hemoglobin; Hct, hematocrit.

**Table 1 medicina-58-00731-t001:** The baseline characteristics of the cases in Analysis 1.

	**Zinc Supplementation**	**No Zinc Supplementation**	** *p* **
	**(*n* = 82)**	**(*n* = 115)**	
Maternal characteristics			
Age (years)	34.0 (30.0–39.0)	34.0 (30.0–37.0)	0.549
Primiparity	43 (52.4)	55 (47.8)	0.523
Infertility treatment	33(40.2)	43 (37.4)	0.685
Body Mass Index	25.0 (22.9–26.9)	25.2 (22.6–27.4)	0.764
Smoking	2 (2.4)	3 (2.6)	0.656
Gestational age at delivery (weeks)	38.0 (36.3–38.0)	37.0 (37.0–38.0)	0.943
Bleeding (g)	954.5 (733.3–1285.0)	850.0 (463.0–1078.0)	0.014
Intravenous Fe treatment	66 (80.5)	34 (29.6)	<0.001
Oral Fe treatment	16 (19.5)	81 (70.4)	<0.001
Dosage of Fe treatment (mg)	410 (240–500)	400 (400–500)	0.280
Gestational Diabetes Mellitus	2 (2.4)	5 (0.43)	0.724
Thyroid dysfunction	4 (4.9)	5(4.3)	1.000
	**Zinc Supplementation**	**No Zinc Supplementation**	** *p* **
	**(*n* = 93)**	**(*n* = 125)**	
Neonatal characteristics			
Male	43 (46.2)	69 (55.2)	0.366
Birth weight (g)	2766 (2245–3064)	2758 (2374–3040)	0.652
5-min Apgar score	9.0 (9.0–9.0)	9.0 (8.0–9.0)	0.917
Cord blood pH	7.34 (7.31–7.35)	7.32 (7.31–7.34)	0.047
Cord blood BE	−1.3 (−2.5–−0.3)	−1.2 (−2.5–−0.3)	0.875

Data are presented as medians (interquartile ranges) or *n* (%).

**Table 2 medicina-58-00731-t002:** The baseline characteristics of the cases in Analysis 2.

	**Zinc Supplementation**	**No Zinc Supplementation**	** *p* **
	**(*n* = 61)**	**(*n* = 87)**	
Maternal characteristics			
Age (years)	34.0(31.0–39.3)	35.0 (30.0–37.0)	0.355
Primiparity	31 (50.8)	43 (49.4)	0.867
Infertility treatment	27 (44.3)	36 (40.9)	0.727
Body Mass Index	25.0 (23.0–26.4)	25.8 (23.8–27.4)	0.114
Smoking	2 (3.3)	3 (3.4)	0.955
Gestational age at delivery (weeks)	38.0 (37.0–38.0)	38.0 (37.0–38.0)	0.889
Gestational Diabetes Mellitus	1 (1.6)	3 (3.4)	0.643
Thyroid dysfunction	2 (3.3)	3 (3.4)	1.000
EPDS score	3 (1–5)	2 (1–6)	0.629
EPDS ≥ 9	3 (4.9)	14 (16.1)	0.023
	**Zinc Supplementation**	**No Zinc Supplementation**	** *p* **
	**(*n* = 70)**	**(*n* = 93)**	
Neonatal characteristics			
Male	35 (50.0)	55 (59.1)	0.245
Birth weight (g)	2956 (2584–3164)	2876 (2590–3095)	0.640
5-min Apgar score	9.0 (9.0–9.0)	9.0 (9.0–9.0)	0.919
Cord blood pH	7.34 (7.31–7.35)	7.33 (7.31–7.34)	0.038
Cord blood BE	−1.4 (−2.5–−0.4)	−1.2 (−2.5–−0.3)	0.674

Data are presented as medians (interquartile ranges) or *n* (%).

**Table 3 medicina-58-00731-t003:** Logistic regression analysis for the risk of EPDS ≥ 9.

	Crude OR	*p*	Adjusted OR	*p*
Age (years)	0.911 (0.828–1.003)	0.057	0.876 (0.780–0.984)	0.025
Primiparity	1.979 (0.691–5.667)	0.204	1.794 (0.582–5.534)	0.309
Infertility treatment	1.604 (0.582–4.422)	0.361	2.480 (0.728–8.449)	0.146
Body Mass Index	0.997 (0.863–1.151)	0.964	0.946 (0.811–1.104)	0.481
Gestational age at delivery (weeks)	0.965 (0.747–1.248)	0.788	0.920 (0.701–1.208)	0.550
Zinc supplementation	0.270 (0.074–0.983)	0.047	0.249 (0.062–0.988)	0.048

Results are reported as odds ratios (95% CI). OR, odds ratio; CI, confidence interval.

## Data Availability

The datasets used and/or analyzed during the current study are available from the corresponding author upon reasonable request.

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
