# Peer review of "The Possible Effects of Zinc Supplementation on Postpartum Depression and Anemia"

_medicina, 2022, doi:10.3390/medicina58060731_

Round 1
Reviewer 1 Report
The authors have provided an interesting study, which suggests that Zn supplementation can improve postpartum depression scores. However, the patients included in these study only gave birth through C section,were Zn deficient and a great percentage of them required intravenous iron supplementation. Therefore, results reported in this study are greatly influenced by selection biases.
- Why did you choose to include only patients who underwent a C-section?
- Results from table 1 should be detailed more; especially the significant differences between the two study groups regarding type of iron supplementation should be clarified.
- The choice of only including patients undergoing C section, as well as only patients with Zn deficiency or marginal Zn deficiency greatly influences the results of the present study. These are selection biases which need to be adressed within the discussion section.
- Lines 133-134: "A statistical difference was found in cord blood pH, but the difference is not considered to be clinically significant." Why was this difference not clinically significant? Please detail.
- Lines 169-170: "Because of the known competition between Zn and Fe, we investigated whether the addition of Zn to Fe-supplementation affected the hematological status." . Please provide the appropriate references for this statement.
- The discussion section should be further expanded. I suggest the authors to refer to the statistically significant results.
- English language requires extensive editing and revision.
Author Response
Thank you for your comments and suggestions that allowed us to greatly improve the quality of the manuscript. We agree with all your comments, and we corrected point by point the manuscript accordingly.
Please see the attachment.

Reviewer 2 Report
The review of the manuscript entitled: “The possible effects of zinc-supplementation on postpartum depression and anemia”
Comments to the authors;
Thank you for the valuable research you have done. The comments would be:
1) In the Introduction section, at the last paragraph, the authors say: “Therefore, our purpose in this study is to investigate the effects of postpartum Zn-supplementation on the Edinburgh Postnatal Depression Scale (EPDS) score as a surrogate endpoint of post-partum depression”. However, the methodology of the study is not match such goal and it has been designed to determine the probability of co-occurrence of Zn deficiency and EPDS scores of 9 or more. Furthermore, the authors have not presented any data about EPDS score changes during weeks after Zn –supplementation in cases who had received Zn. The only data available is about Zn serum levels, 6 days after CS operation.
2) Results section, the last Paragraph, Line 4, the authors say: “Thus, we examined the relationship between Zn-supplementation and an EPDS score of ≥ 9 in 149 cases (Figure 1, Analysis 2)”. The authors, in Abstract and Methods sections also specified that the number for cases who had EPDS data was 149. However, based on Figure 1, Analysis 2 and Table 2; the total number of cases with EPDS data who has received Zn-supplementation (61) and who has not received Zn-supplementation (87) would be 148 (61+87).
3) In Discussion section, paragraph 1, line 2, the authors say: “Our findings suggested a novel clinical implication of postpartum Zn-supplementation; Elevated maternal zinc concentration by postpartum Zn-supplementation may have an attenuating effect on the EPDS score”. As, the authors have not presented any data about EPDS score changes during weeks after Zn –supplementation; the study cannot determine the effect of Zn- supplementation on EPDS scores or depression.
4) In Discussion section, paragraph 1, the last sentence, the authors say: “To the best of our knowledge, this is the first report to evaluate the effects of postpartum Zn-supplementation on post- partum depression and postpartum anemia”. However, by searching literature, you can find RCT studies which has been designed to evaluate such aim. An example: https://pubmed.ncbi.nlm.nih.gov/27617502/
5) In Discussion section, paragraph 2, line 4, the authors say: “These previous reports could support our findings that the postpartum Zn-supplementation has positive effects on the EPDS score”. The same comment as comment number 3 can be considered.
6) In Conclusions section, line 1, the authors say: “We reported that postpartum Zn-supplementation significantly improved the status of maternal blood Zn levels and caused a significant positive improvement for postpartum EPDS score. Our data indicated that postpartum Zn-supplementation could have a possibility to prevent postpartum depression.” The same comment as comment number 3 can be considered.
Good luck
Author Response
We sincerely appreciate your generous comments and helpful suggestions, which are very valuable to improve our manuscript. As you suggested, we have revised the relevant part.
Please see the attachment.

Reviewer 3 Report
The authors investigated whether zinc-supplementation affected the perioperative levels of zinc, hemoglobin, and hematocrit. This is an interesting study but the conclusion is not convincing. My detailed comments are below:
- In another clinical study, zinc supplementation did not improve postnatal depression (PMID: 27617502), the authors need to synthesise the relevant clinical studies, comparing how their own study differs from others, what new conclusions have been reached and what possible reasons exist when there are different conclusions?
- Authors need to include the specific ethics approval number in the methods section
- The original EPDS form used for the study should be included in the supplementary material.
- Too few factors were included in the maternal baseline information, such as the lack of the more common GDM and thyroid function abnormalities.
- In Table 3, the p-value is 0.048, therefore, this study is not very convincing and further inclusion of the sample and refinement of the experimental design is needed.
- The discussion section is too brief and does not summarise the relevant research of the other researchers.
- The conclusion, based on the results of this study, cannot be written with too much certainty.
Author Response
We sincerely appreciate your generous comments and helpful suggestions, which are very valuable to improve our manuscript. We corrected point by point the manuscript accordingly. Our responses to the comments of the reviewer are shown below.
Please see the attachment.

Round 2
Reviewer 1 Report
Please revise English language. Use the help of a professional editing service if needed.
Reviewer 2 Report
The review of the manuscript entitled: “The possible effects of zinc-supplementation on postpartum depression and anemia”
Comments for authors;
Thank you for accurate revision you have done. The paper is now much improved. However, there is some comments for authors to make their study stronger and more attractive for other scientists to refer in their future studies:
1) In the results section, it is recommended for authors to include the mean and standard deviation of EPDS scores in ‘Analyses 2’ results reports for each group of Zn-supplementation (n=61) or no-Zn-supplementation (n=87) and precise number and percent of depressed cases which was detected based on EPDS in each group.
2) In this study, the authors have studied the impact of Zn-supplementation on cases of Anemia who received Fe-treatment simultaneously (based on Figure-1, Analyse 2). Therefore, the results of EPDS in these cases can be highly biased because of ‘Fe’ or both ‘Fe + Zn’ supplementation. This should be included in limitations of the study.
Good luck
Author Response

(The authors gave the same response as above.)

Reviewer 3 Report
The manuscript is improved after revision.
Author Response
We sincerely appreciate the time and effort you have dedicated to providing insightful feedback on ways to strengthen our paper.
This manuscript is a resubmission of an earlier submission. The following is a list of the peer review reports and author responses from that submission.
Round 1
Reviewer 1 Report
My suggestions/queries for the authors are as follows:
- The issue of post-partum depression and zinc status is an important, clinically-relevant topic. This is particularly true given that (non-pregnant) patients with depression have previously been shown to have low serum zinc levels and the severity of the depression is inversely proportional to serum zinc levels. It would be helpful to cite some previous work on the relationship between serum zinc and depression in the introduction.
- The authors cite references 13 and 16 that show a relationship between post-partum depression and iron status. Was this relationship observed in the current data set?
- It would be important to know the interactions between Fe and Zn on EPDS. The authors discuss the interactions between Fe and Zn on hematological status (page 6, beginning on line 164). However, there do not appear to be any statistical analyses of the interactions between Fe and zinc status or supplementation on depression. This seems to be important, as the focus of this paper is post-partum depression.
- Of minor note, in table 1 the units for age (years), weight (g), and bleeding (g) are listed in the table, but there are no units for IV Fe treatment or oral Fe treatment. I found in the text that what is listed is percent. This should be added to the table. Additionally, instead of "bleeding (g)" these data would be better listed as "blood loss (g)".
Author Response
Thank you for your comments and suggestions that allowed us to greatly improve the quality of the manuscript. We agree with all your comments, and we corrected point by point the manuscript accordingly. Our responses to the comments of the reviewer are shown in the attached Word file.

Reviewer 2 Report
- This study well designed and well written.
- This study answered the research question and it contains valuable information regarding the effect of using zinc supplements for the effects of postpartum zinc-supplementation on the Edinburgh Postnatal Depression Scale.
- It concluded that Zinc supplementation had a negative but temporary effect on hematological status in women with postpartum anemia treated with oral iron supplementation and it could be beneficial in postpartum women.
- In line 39-41, this sentence should be separated and corrected as follows;
A systematic review did not show the benefits of Zn-supplementation; A recent 39 Cochrane review reported that Zn-supplementation did not affect pregnancy outcomes 40 including preterm birth, hypertensive disorder in pregnancy, and neonatal outcomes.
- In line 58-60; this sentence needs to be rewritten correctly.
Therefore, we aimed to investigate the effects of postpartum Zn-supplementation on 58 the Edinburgh Postnatal Depression Scale (EPDS) score as a surrogate endpoint of post-59 partum depression. Moreover, we investigated whether adding Zn to Fe-supplementation 60 affected the hematological status of postpartum women.
- References should be prepared according to the journal instructions
- Name of journal should be abbreviated and the year of publication should be written in bold.
Author Response
We sincerely appreciate your generous comments and helpful suggestions, which are very valuable to improve our manuscript. As you suggested, we have revised the relevant part including the style of Reference section.
Reviewer 3 Report
Dear Authors
Dear Editor / Dear authors.
Ensuring the health of the mother and child refers to the tasks of ensuring state security. If the condition of the mother after childbirth can be improved by the appointment of a diet, then you need to use it, but use it with the scientific justification. The topic touched upon in the article is relevant. The scientific content of the manuscript justifies its publication, but some additions and modifications will significantly improve the quality of the article.
Major comments:
1) Introduction, the purpose of the study should be formulated.
2) M&M, when organizing clinical trials with people, they must not only be approved, but also registered.
3) The clinical trial protocol should have been described (which drug Zn and Fe were taken, according to which scheme, which patient indicators were measured, whether the consent of the test patients was obtained, how the presence of postpartum depression was determined, etc.)
4) L.165-167, the reason may be inadequate therapy (the regimen of the Fe and Zn as prescribed incorrectly)
5) Conclusions, conclusions should be reinforced by the authors' comments on the results obtained and the possibility of using them.
6) In the References, 43% of publications refer to 2017-2021 (the last 5 years); the remaining 57% of used sources are older than 5 years. It is recommended to increase the share of references to sources published over the last 5 years when analyzing the current state of research in the area under consideration, since this area of knowledge is rapidly developing.
Author Response
We sincerely appreciate your generous comments and helpful suggestions, which are very valuable to improve our manuscript. We corrected point by point the manuscript accordingly. Our responses to the comments of the reviewer are shown in the attached Word file.

Round 2
Reviewer 3 Report
Dear Authors
My comments are taken into account